# Energy-based Out-of-distribution Detection

**Weitang Liu**
Department of Computer Science and Engineering
University of California, San Diego
La Jolla, CA 92093, USA
`wel022@ucsd.edu`

**Xiaoyun Wang**
Department of Computer Science
University of California, Davis
Davis, CA 95616, USA
`xiywang@ucdavis.edu`

**John D. Owens**
Department of Electrical and Computer Engineering
University of California, Davis
Davis, CA 95616, USA
`jowens@ece.ucdavis.edu`

**Yixuan Li**
Department of Computer Sciences
University of Wisconsin-Madison
Madison, WI 53703, USA
`sharonli@cs.wisc.edu`

## Abstract

Determining whether inputs are out-of-distribution (OOD) is an essential building block for safely deploying machine learning models in the open world. However, previous methods relying on the softmax confidence score suffer from overconfident posterior distributions for OOD data. We propose a unified framework for OOD detection that uses an *energy score*. We show that energy scores better distinguish in- and out-of-distribution samples than the traditional approach using the softmax scores. Unlike softmax confidence scores, energy scores are theoretically aligned with the probability density of the inputs and are less susceptible to the overconfidence issue. Within this framework, energy can be flexibly used as a scoring function for any pre-trained neural classifier as well as a trainable cost function to shape the energy surface explicitly for OOD detection. On a CIFAR-10 pre-trained WideResNet, using the energy score reduces the average FPR (at TPR 95%) by 18.03% compared to the softmax confidence score. With energy-based training, our method outperforms the state-of-the-art on common benchmarks.

## 1 Introduction

The real world is open and full of unknowns, presenting significant challenges for machine learning models that must reliably handle diverse inputs. Out-of-distribution (OOD) uncertainty arises when a machine learning model sees an input that differs from its training data, and thus should not be predicted by the model. Determining whether inputs are out-of-distribution is an essential problem for deploying ML in safety-critical applications such as rare disease identification. A plethora of recent research has studied the issue of out-of-distribution detection [2, 3, 13–15, 19, 22, 23, 26].

Previous approaches rely on the softmax confidence score to safeguard against OOD inputs [13]. An input with a low softmax confidence score is classified as OOD. However, neural networks can produce arbitrarily high softmax confidence for inputs far away from the training data [29]. Such a failure mode occurs since the softmax posterior distribution can have a label-overfitted output space, which makes the softmax confidence score suboptimal for OOD detection.

In this paper, we propose to detect OOD inputs using an *energy score*, and provide both mathematical insights and empirical evidence that the energy score is superior to both a softmax-based score and generative-based methods. The energy-based model [20] maps each input to a single scalar that is lower for observed data and higher for unobserved ones. We show that the energy score is desirable

for OOD detection since it is theoretically aligned with the probability density of the input—samples with higher energies can be interpreted as data with a lower likelihood of occurrence. In contrast, we show mathematically that the softmax confidence score is a biased scoring function that is not aligned with the density of the inputs and hence is not suitable for OOD detection.

Importantly, the energy score can be derived from a *purely discriminative* classification model without relying on a density estimator explicitly, and therefore circumvents the difficult optimization process in training generative models. This is in contrast with JEM [11], which derives the likelihood score $\log p(\mathbf{x})$ from a *generative* modeling perspective. JEM's objective can be intractable and unstable to optimize in practice, as it requires the estimation of the normalized densities over the entire input space to maximize the likelihood. Moreover, while JEM only utilizes in-distribution data, our framework allows exploiting both the in-distribution and the auxiliary outlier data to shape the energy gap flexibly between the training and OOD data, a learning method that is much more effective than JEM or Outlier Exposure [14].

**Contributions.** We propose a unified framework using an energy score for OOD detection.[1] We show that one can flexibly use energy as both a *scoring function* for any pre-trained neural classifier (without re-training), and a *trainable cost function* to fine-tune the classification model. We demonstrate the effectiveness of energy function for OOD detection for both use cases.

- At *inference time*, we show that energy can conveniently replace softmax confidence for any pre-trained neural network. We show that the energy score outperforms the softmax confidence score [13] on common OOD evaluation benchmarks. For example, on WideResNet, the energy score reduces the average FPR (at 95% TPR) by **18.03**% on CIFAR-10 compared to using the softmax confidence score. Existing approaches using pre-trained models may have several hyperparameters to be tuned and sometimes require additional data. In contrast, the energy score is a parameter-free measure, which is easy to use and implement, and in many cases, achieves comparable or even better performance.
- At *training time*, we propose an energy-bounded learning objective to fine-tune the network. The learning process shapes the energy surface to assign low energy values to the in-distribution data and higher energy values to OOD training data. Specifically, we regularize the energy using two square hinge loss terms, which explicitly create the energy gap between in- and out-of-distribution training data. We show that the energy fine-tuned model outperforms the previous state-of-the-art method evaluated on six OOD datasets. Compared to the softmax-based fine-tuning approach [14], our method reduces the average FPR (at 95% TPR) by **10.55**% on CIFAR-100. This fine-tuning leads to improved OOD detection performance while maintaining similar classification accuracy on in-distribution data.

The rest of the paper is organized as follows. Section 2 provides the background of energy-based models. In Section 3, we present our method of using energy score for OOD detection, and experimental results in Section 4. Section 5 provides an comprehensive literature review on OOD detection and energy-based learning. We conclude in Section 6, with discussion on broader impact in Section 7.

## 2 Background: Energy-based Models

The essence of the energy-based model (EBM) [20] is to build a function $E(\mathbf{x}) : \mathbb{R}^D \to \mathbb{R}$ that maps each point $\mathbf{x}$ of an input space to a single, non-probabilistic scalar called the *energy*. A collection of energy values could be turned into a probability density $p(\mathbf{x})$ through the Gibbs distribution:

$$p(y \mid \mathbf{x}) = \frac{e^{-E(\mathbf{x},y)/T}}{\int_{y'} e^{-E(\mathbf{x},y')/T}} = \frac{e^{-E(\mathbf{x},y)/T}}{e^{-E(\mathbf{x})/T}}, \tag{1}$$

where the denominator $\int_{y'} e^{-E(\mathbf{x},y')/T}$ is called the partition function, which marginalizes over $y$, and $T$ is the temperature parameter. The *Helmholtz free energy* $E(\mathbf{x})$ of a given data point $\mathbf{x} \in \mathbb{R}^D$ can be expressed as the negative of the log partition function:

$$E(\mathbf{x}) = -T \cdot \log \int_{y'} e^{-E(\mathbf{x},y')/T} \tag{2}$$

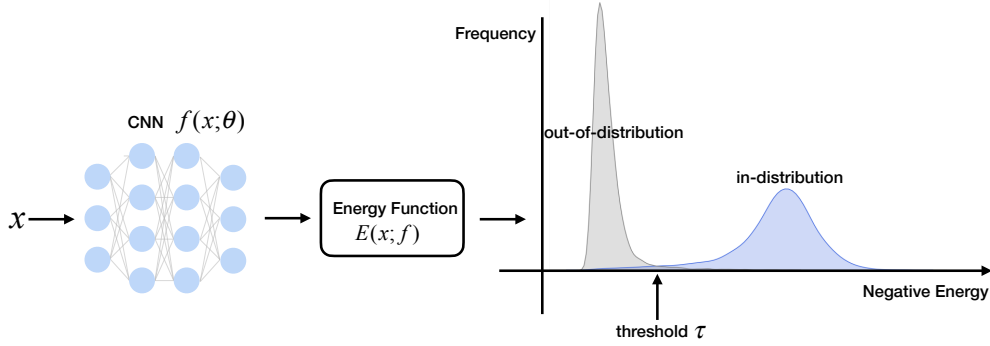

Figure 1: Energy-based out-of-distribution detection framework. The energy can be used as a scoring function for any pre-trained neural network (without re-training), or used as a trainable cost function to fine-tune the classification model. During inference time, for a given input $\mathbf{x}$, the energy score $E(\mathbf{x}; f)$ is calculated for a neural network $f(\mathbf{x})$. The OOD detector will classify the input as OOD if the negative energy score is smaller than the threshold value.

**Energy Function** The energy-based model has an inherent connection with modern machine learning, especially discriminative models. To see this, we consider a discriminative neural classifier $f(\mathbf{x})$ : $\mathbb{R}^D \to \mathbb{R}^K$, which maps an input $\mathbf{x} \in \mathbb{R}^D$ to $K$ real-valued numbers known as logits. These logits are used to derive a categorical distribution using the softmax function:

$$p(y \mid \mathbf{x}) = \frac{e^{f_y(\mathbf{x})/T}}{\sum_i^K e^{f_i(\mathbf{x})/T}}, \tag{3}$$

where $f_y(\mathbf{x})$ indicates the $y^{\text{th}}$ index of $f(\mathbf{x})$, i.e., the logit corresponding to the $y^{\text{th}}$ class label.

By connecting Eq. 1 and Eq. 3, we can define an energy for a given input $(\mathbf{x}, y)$ as $E(\mathbf{x}, y) = -f_y(\mathbf{x})$. More importantly, without changing the parameterization of the neural network $f(\mathbf{x})$, we can express the free energy function $E(\mathbf{x}; f)$ over $\mathbf{x} \in \mathbb{R}^D$ in terms of the denominator of the softmax activation:

$$E(\mathbf{x}; f) = -T \cdot \log \sum_i^K e^{f_i(\mathbf{x})/T}. \tag{4}$$

## 3 Energy-based Out-of-distribution Detection

We propose a unified framework using an energy score for OOD detection, where the differences of energies between in- and out-of-distribution allow effective differentiation. The energy score mitigates a critical problem of softmax confidence with arbitrarily high values for OOD examples [12]. In the following, we first describe using energy as an OOD score for pre-trained models, and the connection between the energy and softmax scores (Section 3.1). We then describe how to use energy as a trainable cost function for model fine-tuning (Section 3.2).

### 3.1 Energy as Inference-time OOD Score

Out-of-distribution detection is a binary classification problem that relies on a score to differentiate between in- and out-of-distribution examples. A scoring function should produce values that are distinguishable between in- and out-of-distribution. A natural choice is to use the density function of the data $p(\mathbf{x})$ and consider examples with low likelihood to be OOD. While it is possible to obtain the density function for a discriminative model by resorting to the energy-based model [11,20]:

$$p(\mathbf{x}) = \frac{e^{-E(\mathbf{x}; f)/T}}{\int_{\mathbf{x}} e^{-E(\mathbf{x}; f)/T}}, \tag{5}$$

the normalized densities $Z = \int_{\mathbf{x}} e^{-E(\mathbf{x}; f)/T}$ (with respect to $\mathbf{x}$) can be intractable to compute or even reliably estimate over the input space.

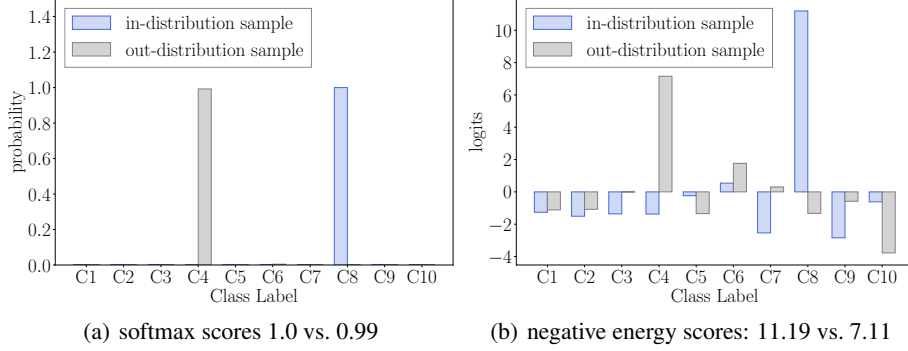

(a) softmax scores 1.0 vs. 0.99          (b) negative energy scores: 11.19 vs. 7.11

Figure 2: (a) Softmax and (b) logit outputs of two samples calculated on a CIFAR-10 pre-trained WideResNet. The out-of-distribution sample is from SVHN. For (a), the softmax confidence scores are 1.0 and 0.99 for the in- and out-of-distribution examples. In contrast, the energy scores calculated from logit are $E(\mathbf{x}_{\text{in}}) = -11.19$, $E(\mathbf{x}_{\text{out}}) = -7.11$. While softmax confidence scores are almost identical for in- and out-distribution samples, energy scores provide more meaningful information with which to differentiate them.

To mitigate the challenge, our key observation is that the absence of the normalization does not affect the OOD detection at all. A data point with a higher probability of occurrence is equivalent to having lower energy. To see this, we can take the logarithm of both sides of Eq. 5,

$$\log p(\mathbf{x}) = -E(\mathbf{x}; f)/T - \underbrace{\log Z}_{\text{constant for all } \mathbf{x}}.$$

The equation above suggests that $-E(\mathbf{x}; f)$ is in fact linearly aligned with the log likelihood function, which is desirable for OOD detection. Examples with higher energies (lower likelihood) are considered as OOD inputs. Specifically, we propose using the energy function $E(\mathbf{x}; f)$ in Eq. 4 for OOD detection:

$$G(\mathbf{x}; \tau, f) = \begin{cases} 0 & \text{if} -E(\mathbf{x}; f) \leq \tau, \\ 1 & \text{if} -E(\mathbf{x}; f) > \tau, \end{cases} \tag{6}$$

where $\tau$ is the energy threshold. In practice, we choose the threshold using in-distribution data so that a high fraction of inputs are correctly classified by the OOD detector $G(\mathbf{x})$. Here we use negative energy scores, $-E(\mathbf{x}; f)$, to align with the conventional definition where positive (in-distribution) samples have higher scores. The energy score is non-probabilistic in nature, which can be conveniently calculated via the `logsumexp` operator. Unlike JEM [11], our method does not require estimating the density $Z$ explicitly, as $Z$ is sample-independent and does not affect the overall energy score distribution.

**Energy Score vs. Softmax Score** Our method can be used as a simple and effective replacement for the softmax confidence score [13] for any pre-trained neural network. To see this, we first derive a mathematical connection between the energy score and the softmax confidence score:

$$\max_y p(y \mid \mathbf{x}) = \max_y \frac{e^{f_y(\mathbf{x})}}{\sum_i e^{f_i(\mathbf{x})}} = \frac{e^{f^{\max}(\mathbf{x})}}{\sum_i e^{f_i(\mathbf{x})}}$$

$$= \frac{1}{\sum_i e^{f_i(\mathbf{x}) - f^{\max}(\mathbf{x})}}$$

$$\implies \log \max_y p(y \mid \mathbf{x}) = E(\mathbf{x}; f(\mathbf{x}) - f^{\max}(\mathbf{x})) = E(\mathbf{x}; f) + f^{\max}(\mathbf{x}),$$

when $T = 1$. This reveals that the log of the softmax confidence score is in fact equivalent to a special case of the free energy score, where all the logits are shifted by their maximum logit value. Since $f^{\max}(\mathbf{x})$ tends to be higher and $E(\mathbf{x}; f)$ tends to be lower for in-distribution data, the shifting results in a biased scoring function that is no longer proportional to the probability density $p(\mathbf{x})$ for $\mathbf{x} \in \mathbb{R}^D$:

$$\log \max_y p(y \mid \mathbf{x}) = -\log p(\mathbf{x}) + \underbrace{f^{\max}(\mathbf{x}) - \log Z}_{\text{Not constant. Larger for in-dist } \mathbf{x}}$$

$$\not\propto -\log p(\mathbf{x}).$$

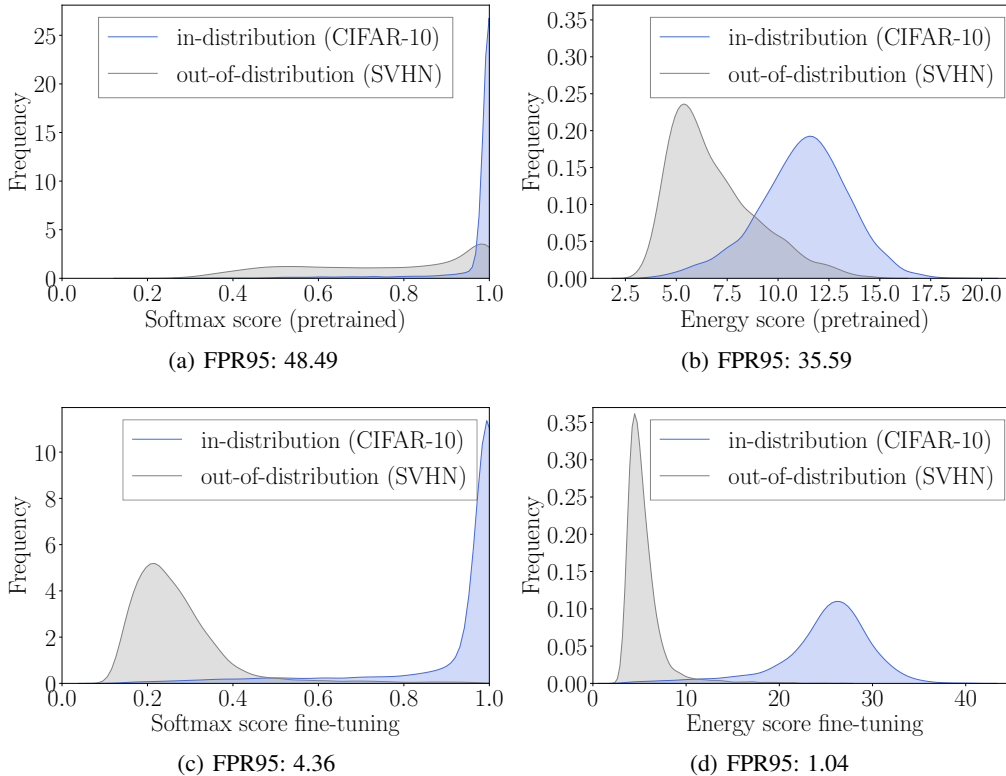

Figure 3: (a & b) Distribution of softmax scores vs. energy scores from pre-trained WideResNet. We contrast the score distribution from fine-tuned models using Outlier Exposure [14] (c) and our energy-bounded learning (d). We use negative energy scores for (b & d) to align with the convention that positive (in-distribution) samples have higher scores. Using energy score leads to an overall smoother distribution (b & d), and is less susceptible to the spiky distribution that softmax exhibits for in-distribution data (a & c).

As a result, unlike the energy score, which is well aligned with density $p(\mathbf{x})$, the softmax confidence score is less able to reliably distinguish in- and out-of-distribution examples. To illustrate with a real example, Figure 2 shows one example from the SVHN dataset (OOD) and another example from the in-distribution data CIFAR-10. While their softmax confidence scores are almost identical (1.0 vs 0.99), the negative energy scores are more distinguishable (11.19 vs. 7.11). Thus, working in the original logit space (energy score) instead of the shifted logit space (softmax score) yields more useful information for each sample. We show in our experimental results in Section 4.2 that energy score is a superior metric for OOD detection than the softmax score.

## 3.2 Energy-bounded Learning for OOD Detection

While energy score can be useful for a pre-trained neural network, the energy gap between in- and out-of-distribution might not always be optimal for differentiation. Therefore, we also propose an energy-bounded learning objective, where the neural network is fine-tuned to *explicitly* create an energy gap by assigning lower energies to the in-distribution data, and higher energies to the OOD data. The learning process allows greater flexibility in contrastively shaping the energy surface, resulting in more distinguishable in- and out-of-distribution data. Specifically, our energy-based classifier is trained using the following objective:

$$\min_{\theta} \quad \mathbb{E}_{(\mathbf{x},y) \sim \mathcal{D}_{\text{in}}^{\text{train}}}[-\log F_y(\mathbf{x})] + \lambda \cdot L_{\text{energy}} \tag{7}$$

where $F(\mathbf{x})$ is the softmax output of the classification model and $\mathcal{D}_{\text{in}}^{\text{train}}$ is the in-distribution training data. The overall training objective combines the standard cross-entropy loss, along with a

| $\mathcal{D}_{\text{in}}^{\text{test}}$ | fine-tune? | OOD dataset $\mathcal{D}_{\text{out}}^{\text{test}}$ | FPR95 ↓ | AUROC ↑ | AUPR ↑ |
|---|---|---|---|---|---|
| | | | Softmax score [13] / Energy score (ours) | | |
| **WideResNet** CIFAR-10 | ✗ | iSUN | 56.03 / **33.68** | 89.83 / **92.62** | 97.74 / **98.27** |
| | | Places365 | 59.48 / **40.14** | 88.20 / **89.89** | 97.10 / **97.30** |
| | | Texture | 59.28 / **52.79** | **88.50** / 85.22 | **97.16** / 95.41 |
| | | SVHN | 48.49 / **35.59** | **91.89** / 90.96 | **98.27** / 97.64 |
| | | LSUN-Crop | 30.80 / **8.26** | 95.65 / **98.35** | 99.13 / **99.66** |
| | | LSUN-Resize | 52.15 / **27.58** | 91.37 / **94.24** | 98.12 / **98.67** |
| | | **average** | 51.04 / **33.01** | 90.90 / **91.88** | **97.92** / 97.83 |
| | | | OE fine-tune [14] / Energy fine-tune (ours) | | |
| **WideResNet** CIFAR-10 | ✓ | iSUN | 6.32 / **1.60** | 98.85 / **99.33** | 99.77 / **99.87** |
| | | Places365 | 19.07 / **9.00** | 96.16 / **97.48** | 99.06 / **99.35** |
| | | Texture | 12.94 / **5.34** | 97.73 / **98.56** | 99.52 / **99.68** |
| | | SVHN | 4.36 / **1.04** | 98.63 / **99.41** | 99.74 / **99.89** |
| | | LSUN-Crop | 2.89 / **1.67** | **99.49** / 99.32 | **99.90** / 99.86 |
| | | LSUN-Resize | 5.59 / **1.25** | 98.94 / **99.39** | 99.79 / **99.88** |
| | | **average** | 8.53 / **3.32** | 98.30 / **98.92** | 99.63 / **99.75** |

Table 1: OOD detection performance comparison using softmax-based vs. energy-based approaches. We use WideResNet [47] to train on the in-distribution dataset CIFAR-10. We show results for both using the pretrained model (top) and applying fine-tuning (bottom). All values are percentages. ↑ indicates larger values are better, and ↓ indicates smaller values are better. **Bold** numbers are superior results.

regularization loss defined in terms of energy:

$$L_{\text{energy}} = \mathbb{E}_{(\mathbf{x}_{\text{in}}, y) \sim \mathcal{D}_{\text{in}}^{\text{train}}} (\max(0, E(\mathbf{x}_{\text{in}}) - m_{\text{in}}))^2 \qquad (8)$$

$$+ \mathbb{E}_{\mathbf{x}_{\text{out}} \sim \mathcal{D}_{\text{out}}^{\text{train}}} (\max(0, m_{\text{out}} - E(\mathbf{x}_{\text{out}})))^2 \qquad (9)$$

where $\mathcal{D}_{\text{out}}^{\text{train}}$ is the unlabeled auxiliary OOD training data [38]. In particular, we regularize the energy using two squared hinge loss terms[2] with separate margin hyperparameters $m_{\text{in}}$ and $m_{\text{out}}$. In one term, the model penalizes in-distribution samples that produce energy higher than the specified margin parameter $m_{\text{in}}$. Similarly, in another term, the model penalizes the out-of-distribution samples with energy lower than the margin parameter $m_{\text{out}}$. In other words, the loss function penalizes the samples with energy $E(\mathbf{x}) \in [m_{\text{in}}, m_{\text{out}}]$. Once the model is fine-tuned, the downstream OOD detection is similar to our description in Section 3.1.

## 4 Experimental Results

In this section, we describe our experimental setup (Section 4.1) and demonstrate the effectiveness of our method on a wide range of OOD evaluation benchmarks. We also conduct an ablation analysis that leads to an improved understanding of our approach (Section 4.2).

### 4.1 Setup

**In-distribution Datasets** We use the SVHN [28], CIFAR-10 [18], and CIFAR-100 [18] datasets as in-distribution data. We use the standard split, and denote the training and test set by $D_{\text{in}}^{\text{train}}$ and $D_{\text{in}}^{\text{test}}$, respectively.

**Out-of-distribution Datasets** For the OOD test dataset $\mathcal{D}_{\text{out}}^{\text{test}}$, we use six common benchmarks: `Textures` [5], `SVHN` [28], `Places365` [49], `LSUN-Crop` [46], `LSUN-Resize` [46], and `iSUN` [45]. The pixel values of all the images are normalized through z-normalization in which the parameters are dependent on the network type. For the auxiliary outlier dataset, we use 80 Million Tiny Images [38], which is a large-scale, diverse dataset scraped from the web. We remove all examples in this dataset that appear in CIFAR-10 and CIFAR-100.

| $\mathcal{D}_{\text{in}}^{\text{test}}$ | Method | FPR95 | AUROC | AUPR | In-dist Test Error |
|---|---|---|---|---|---|
| | | ↓ | ↑ | ↑ | ↓ |
| **CIFAR-10** (WideResNet) | Softmax score [13] | 51.04 | 90.90 | 97.92 | 5.16 |
| | Energy score (ours) | 33.01 | 91.88 | 97.83 | 5.16 |
| | ODIN [23] | 35.71 | 91.09 | 97.62 | 5.16 |
| | Mahalanobis [22] | 37.08 | 93.27 | 98.49 | 5.16 |
| | OE [14] | 8.53 | 98.30 | 99.63 | 5.32 |
| | Energy fine-tuning (ours) | **3.32** | **98.92** | **99.75** | **4.87** |
| **CIFAR-100** (WideResNet) | Softmax score [13] | 80.41 | 75.53 | 93.93 | **24.04** |
| | Energy score (ours) | 73.60 | 79.56 | 94.87 | 24.04 |
| | ODIN [23] | 74.64 | 77.43 | 94.23 | 24.04 |
| | Mahalanobis [22] | 54.04 | 84.12 | 95.88 | 24.04 |
| | OE [14] | 58.10 | 85.19 | 96.40 | 24.30 |
| | Energy fine-tuning (ours) | **47.55** | **88.46** | **97.10** | 24.58 |

Table 2: Comparison with discriminative-based OOD detection methods. ↑ indicates larger values are better, and ↓ indicates smaller values are better. All values are percentages and are averaged over the six OOD test datasets described in section 4.1. **Bold** numbers are superior results. Detailed results for each OOD test dataset can be found in Appendix A.

**Evaluation Metrics** We measure the following metrics: (1) the false positive rate (FPR95) of OOD examples when true positive rate of in-distribution examples is at 95%; (2) the area under the receiver operating characteristic curve (AUROC); and (3) the area under the precision-recall curve (AUPR).

**Training Details** We use WideResNet [47] to train the image classification models. For energy fine-tuning, the weight $\lambda$ of $L_{\text{energy}}$ is 0.1. We use the same training setting as in Hendryks et al. [14], where the number of epochs is 10, the initial learning rate is 0.001 with cosine decay [24], and the batch size is 128 for in-distribution data and 256 for unlabeled OOD training data. We use the validation set as in Hendrycks et al. [14] to determine the hyperparameters: $m_{\text{in}}$ is chosen from $\{-3, -5, -7\}$, and $m_{\text{out}}$ is chosen from $\{-15, -19, -23, -27\}$ that minimize FPR95. The ranges of $m_{\text{in}}$ and $m_{\text{out}}$ can be chosen around the mean of energy scores from a pre-trained model for in- and out-of-distribution samples respectively. We provide the optimal margin parameters in Appendix B.

## 4.2 Results

**Does energy-based OOD detection work better than the softmax-based approach?** We begin by assessing the improvement of energy score over the softmax score. Table 1 contains a detailed comparison for CIFAR-10. For inference-time OOD detection (without fine-tuning), we compare with the softmax confidence score baseline [13]. We show that using energy score reduces the average FPR95 by **18.03%** compared to the baseline on CIFAR-10. Additional results on SVHN as in-distribution data are provided in Table 6, where we show the energy score consistently outperforms the softmax score by **8.69**% (FPR95).

We also consider energy fine-tuning and compare with Outlier Exposure (OE) [14], which regularizes the softmax probabilities to be uniform distribution for outlier training data. For both approaches, we fine-tune on the same data and use the same training configurations in terms of learning rate and batch size. Our energy fine-tuned model reduces the FPR95 by **5.20%** on CIFAR-10 compared to OE. The improvement is more pronounced on complex datasets such as CIFAR-100, where we show a **10.55%** improvement over OE.

To gain further insights, we compare the energy score distribution for in- and out-of-distribution data. Figure 3 compares the energy and softmax score histogram distributions, derived from pre-trained as well as fine-tuned networks. The energy scores calculated from a pre-trained network on both training and OOD data naturally form smooth distributions (see Figure 3(b)). In contrast, softmax scores for both in- and out-of-distribution data concentrate on high values, as shown in Figure 3(a). Overall our experiments show that using energy makes the scores more distinguishable between in- and out-of-distributions, and as a result, enables more effective OOD detection.

**How does our approach compare to competitive OOD detection methods?** In Table 2, we compare our work against discriminative OOD detection methods that are competitive in literature. All the numbers reported are averaged over six OOD test datasets. We provide detailed results for

| $\mathcal{D}_{\text{in}}^{\text{test}}$ | Method | pre-trained? | SVHN | CIFAR-100 | CelebA |
|---|---|---|---|---|---|
| | Class-conditional Glow [17] | ✗ | 0.64 | 0.65 | 0.54 |
| | IGEBM [8] | ✗ | 0.43 | 0.54 | 0.69 |
| CIFAR-10 | JEM-softmax [11] | ✗ | 0.89 | 0.87 | 0.79 |
| | JEM-likelihood [11] | ✗ | 0.67 | 0.67 | 0.75 |
| | Energy score (ours) | ✓ | **0.91** | **0.87** | **0.78** |
| | Energy fine-tuning (ours) | ✗ | **0.99** | **0.94** | **1.00** |

Table 3: Comparison with generative-based models for OOD detection. Values are AUROC.

each dataset in Appendix A. We note that existing approaches using a pre-trained model have several hyperparameters that need to be tuned, sometimes with the help of additional data and a classifier to be trained (such as Mahalanobis [22]). In contrast, using an energy score on a pre-trained network is parameter-free, easy to use and deploy, and in many cases, achieves comparable or even better performance.

In Table 3, we also compare with state-of-the-art hybrid models that incorporated generative modeling [8, 11, 17]. These approaches are stronger baselines than pure generative-modeling-based OOD detection methods [4, 27, 32], due to the use of labeling information during training. In both cases (with and without fine-tuning), our energy-based method outperforms hybrid models.

**How does temperature scaling affect the energy-based OOD detector?** Previous work ODIN [23] showed both empirically and theoretically that temperature scaling improves out-of-distribution detection. Inspired by this, we also evaluate how the temperature parameter $T$ affects the performance of our energy-based detector. Applying a temperature $T > 1$ rescales the logit vector $f(\mathbf{x})$ by $1/T$. Figure 4 in Appendix A shows how the FPR95 changes as we increase the temperature from $T = 1$ to $T = 1000$. Interestingly, using larger $T$ leads to more uniformly distributed predictions and makes the energy scores less distinguishable between in- and out-of-distribution examples. Our result means that the energy score can be used parameter-free by simply setting $T = 1$.

**How do the margin parameters affect the performance?** Figure 4(b) shows how the performance of energy fine-tuning (measured by FPR) changes with different margin parameters of $m_{\text{in}}$ and $m_{\text{out}}$ on WideResNet. Overall the method is not very sensitive to $m_{\text{out}}$ in the range chosen. As expected, imposing too small of an energy margin $m_{\text{in}}$ for in-distribution data may lead to difficulty in optimization and degradation in performance.

**Does energy fine-tuning affect the classification accuracy of the neural network?** For the inference-time use case, our method does not change the parameters of the pre-trained neural network $f(\mathbf{x})$ and preserves its accuracy. For energy fine-tuned models, we compare classification accuracy of $f(\mathbf{x})$ with other methods in Table 2. When trained on WideResNet with CIFAR-10 as in-distribution, our energy fine-tuned model achieves a test error of 4.98% on CIFAR-10, compared to the OE fine-tuned model's 5.32% and the pre-trained model's 5.16%. Overall this fine-tuning leads to improved OOD detection performance while maintaining almost comparable classification accuracy on in-distribution data.

## 5 Related Work

**Out-of-distribution uncertainty for pre-trained models** The softmax confidence score has become a common baseline for OOD detection [13]. A theoretical investigation [12] shows that neural networks with ReLU activation can produce arbitrarily high softmax confidence for OOD inputs. Several works attempt to improve the OOD uncertainty estimation by using deep ensembles [19], the ODIN score [23], the Mahalanobis distance [22], and generalized ODIN score [15]. DeVries and Taylor [6] propose to learn the confidence score by attaching an auxiliary branch to a pre-trained classifier and deriving an OOD score. However, previous methods are either computationally expensive or require tuning many hyper-parameters. In contrast, in our work, the energy score can be used as a parameter-free measurement, which is easy to use in an OOD-agnostic setting.

**Out-of-distribution detection with model fine-tuning** While it is impossible to anticipate the exact OOD test distribution, previous methods have explored using artificially synthesized data from GANs [21] or unlabeled data [14] as auxiliary OOD training. Auxiliary data allows the model to be explicitly regularized through fine-tuning, producing lower confidence on anomalous examples [2, 9, 25, 26, 36]. A loss function is used to force the predictive distribution of OOD

samples toward uniform distribution [14, 21]. Recently, Mohseni et al. [26] explore training by adding additional background classes for OOD score. Chen et al. [3] propose informative outlier mining by selectively training on auxiliary OOD data that induces uncertain OOD scores, which improves the OOD detection performance on both clean and perturbed adversarial OOD inputs. In our work, we instead regularize the network to produce higher energy on anomalous inputs. Our approach does not alter the semantic class space and can be used both with and without auxiliary OOD data.

**Generative Modeling Based Out-of-distribution Detection.** Generative models [7, 16, 33, 37, 39] can be alternative approaches for detecting OOD examples, as they directly estimate the in-distribution density and can declare a test sample to be out-of-distribution if it lies in the low-density regions. However, as shown by Nalisnick et al. [27], deep generative models can assign a high likelihood to out-of-distribution data. Deep generative models can be more effective for out-of-distribution detection using improved metrics [4], including the likelihood ratio [32, 35]. Though our work is based on discriminative classification models, we show that energy scores can be theoretically interpreted from a data density perspective. More importantly, generative-based models can be prohibitively challenging to train and optimize, especially on large and complex datasets. In contrast, our method relies on a discriminative classifier, which can be much easier to optimize using standard SGD. Our method therefore inherits the merits of generative-based approaches, while circumventing the difficult optimization process in training generative models.

**Energy-based learning** Energy-based machine learning models date back to Boltzmann machines [1, 34], networks of units with an energy defined for the overall network. Energy-based learning [20, 30, 31] provides a unified framework for many probabilistic and non-probabilistic approaches to learning. Recent work [48] also demonstrated using energy functions to train GANs [10], where the discriminator uses energy values to differentiate between real and generated images. Xie et al. [41] first showed that a generative random field model can be derived from a discriminative neural networks. In subsequent works, Xie et al. [40, 42–44] explored using EBMs for video generation and 3D shape pattern generation. While Grathwohl et al. [11] explored using JEM for OOD detection, their optimization objective estimates the joint distribution $p(\mathbf{x}, y)$ from a generative perspective; they use standard probabilistic scores in downstream OOD detection. In contrast, our training objective is purely discriminative, and we show that non-probabilistic energy scores can be directly used as a scoring function for OOD detection. Moreover, JEM requires estimating the normalized densities, which can be challenging and unstable to compute. In contrast, our formulation does not require proper normalization and allows greater flexibility in optimization. Perhaps most importantly, our training objective directly optimizes for the energy gap between in- and out-of-distribution, which fits naturally with the proposed OOD detector that relies on energy score.

# 6 Conclusion and Outlook

In this work, we propose an energy-based framework for out-of-distribution detection. We show that energy score is a simple and promising replacement of the softmax confidence score. The key idea is to use a non-probabilistic energy function that attributes lower values to in-distribution data and higher values to out-of-distribution data. Unlike softmax confidence scores, the energy scores are provably aligned with the density of inputs, and as a result, yield substantially improved OOD detection performance. For future work, we would like to explore using energy-based OOD detection beyond image classification tasks. Our approach can be valuable to other machine learning tasks such as active learning. We hope future research will increase the attention toward a broader view of OOD uncertainty estimation from an energy-based perspective.

# 7 Broader Impact

Our project aims to improve the dependability and trustworthiness of modern machine learning models. This stands to benefit a wide range of fields and societal activities. We believe out-of-distribution uncertainty estimation is an increasingly critical component of systems that range from consumer and business applications (e.g., digital content understanding) to transportation (e.g., driver assistance systems and autonomous vehicles), and to health care (e.g., rare disease identification). Through this work and by releasing our code, we hope to provide machine learning researchers a new methodological perspective and offer machine learning practitioners an easy-to-use tool that renders safety against anomalies in the open world. While we do not anticipate any negative consequences to our work, we hope to continue to improve and build on our framework in future work.

## Acknowledgement

The research at UC Davis was supported by an NVIDIA gift and their donation of a DGX Station. Research at UW-Madison is partially supported by the Office of the Vice Chancellor for Research and Graduate Education with funding from the Wisconsin Alumni Research Foundation (WARF).

## Footnotes

[1]Our code is publicly available to facilitate reproducible research: https://github.com/wetliu/energy_ood.

[2] We also explored using a hinge loss such as $\max(0, E(\mathbf{x}_{\text{in}}) - E(\mathbf{x}_{\text{out}}) + m)$ through a single constant margin parameter $m$. While the difference between $E(\mathbf{x}_{\text{in}})$ and $E(\mathbf{x}_{\text{out}})$ can be stable, their values do not stabilize. Optimization is more flexible and the training process is more stable with two separate hinge loss terms.

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
