[Supplementary Material]

# Supplementary Material:

## Energy-based Out-of-distribution Detection

## A  Detailed Experimental Results

We report the performance of OOD detectors on each of the six OOD test datasets in Table 4 (CIFAR-10) and Table 5 (CIFAR-100).

| Dataset $\mathcal{D}_{out}^{test}$ | | FPR95 ↓ | AUROC ↑ | AUPR ↑ |
|---|---|---|---|---|
| **Texture** | Softmax score [14] | 59.28 | 88.50 | 97.16 |
| | Energy score (ours) | 52.79 | 85.22 | 95.41 |
| | ODIN [24] | 49.12 | 84.97 | 95.28 |
| | Mahalanobis [23] | 15.00 | 97.33 | 99.41 |
| | OE [15] | 12.94 | 97.73 | 99.52 |
| | Energy fine-tuning (ours) | **5.34** | **98.56** | **99.68** |
| **SVHN** | Softmax score [14] | 48.49 | 91.89 | 98.27 |
| | Energy score (ours) | 35.59 | 90.96 | 97.64 |
| | ODIN [24] | 33.55 | 91.96 | 98.00 |
| | Mahalanobis [23] | 12.89 | 97.62 | 99.47 |
| | OE [15] | 4.36 | 98.63 | 99.74 |
| | Energy fine-tuning (ours) | **1.04** | **99.41** | **99.89** |
| **Places365** | Softmax score [14] | 59.48 | 88.20 | 97.10 |
| | Energy score (ours) | 40.14 | 89.89 | 97.30 |
| | ODIN [24] | 57.40 | 84.49 | 95.82 |
| | Mahalanobis [23] | 68.57 | 84.61 | 96.20 |
| | OE [15] | 19.07 | 96.16 | 99.06 |
| | Energy fine-tuning (ours) | **9.00** | **97.48** | **99.35** |
| **LSUN-C** | Softmax score [14] | 30.80 | 95.65 | 99.13 |
| | Energy score (ours) | 8.26 | 98.35 | 99.66 |
| | ODIN [24] | 15.52 | 97.04 | 99.33 |
| | Mahalanobis [23] | 39.22 | 94.15 | 98.81 |
| | OE [15] | 2.89 | **99.49** | **99.90** |
| | Energy fine-tuning (ours) | **1.67** | 99.32 | 99.86 |
| **LSUN Resize** | Softmax score [14] | 52.15 | 91.37 | 98.12 |
| | Energy score (ours) | 27.58 | 94.24 | 98.67 |
| | ODIN [24] | 26.62 | 94.57 | 98.77 |
| | Mahalanobis [23] | 42.62 | 93.23 | 98.60 |
| | OE [15] | 5.59 | 98.94 | 99.79 |
| | Energy fine-tuning (ours) | **1.25** | **99.39** | **99.88** |
| **iSUN** | Softmax score [14] | 56.03 | 89.83 | 97.74 |
| | Energy score (ours) | 33.68 | 92.62 | 98.27 |
| | ODIN [24] | 32.05 | 93.50 | 98.54 |
| | Mahalanobis [23] | 44.18 | 92.66 | 98.45 |
| | OE [15] | 6.32 | 98.85 | 99.77 |
| | Energy fine-tuning (ours) | **1.60** | **99.33** | **99.87** |

Table 4: OOD Detection performance of CIFAR-10 as in-distribution for each OOD test dataset. The Mahalanobis score is calculated using the features of the second-to-last layer. **Bold** numbers are superior results.

## B  Details of Experiments

**Software and Hardware.** We run all experiments with PyTorch and NVIDIA Tesla V100 DGXS GPUs.

**Number of Evaluation Runs.** We fine-tune the models once with a fixed random seed. Following OE [15], reported performance for each OOD dataset is averaged over 10 random batches of samples.

**Average Runtime** On a single GPU, the running time for energy fine-tuning is around 6 minutes; each training epoch takes 34 seconds. The evaluation time for all six OOD datasets is approximately 4 minutes.

| Dataset $\mathcal{D}_{\text{out}}^{\text{test}}$ | | FPR95 $\downarrow$ | AUROC $\uparrow$ | AUPR $\uparrow$ |
|---|---|---|---|---|
| **Texture** | Softmax score [14] | 83.29 | 73.34 | 92.89 |
| | Energy score (ours) | 79.41 | 76.28 | 93.63 |
| | ODIN [24] | 79.27 | 73.45 | 92.75 |
| | Mahalanobis [23] | **39.39** | **90.57** | **97.74** |
| | OE [15] | 61.11 | 84.56 | 96.19 |
| | Energy fine-tuning (ours) | 57.01 | 87.40 | 96.95 |
| **SVHN** | Softmax score [14] | 84.59 | 71.44 | 92.93 |
| | Energy score (ours) | 85.82 | 73.99 | 93.65 |
| | ODIN [24] | 84.66 | 67.26 | 91.38 |
| | Mahalanobis [23] | 57.52 | 86.01 | 96.68 |
| | OE [15] | 65.91 | 86.66 | 97.09 |
| | Energy fine-tuning (ours) | **28.97** | **95.40** | **99.05** |
| **Places365** | Softmax score [14] | 82.84 | 73.78 | 93.29 |
| | Energy score (ours) | 80.56 | 75.44 | 93.45 |
| | ODIN [24] | 87.88 | 71.63 | 92.56 |
| | Mahalanobis [23] | 88.83 | 67.87 | 90.71 |
| | OE [15] | 57.92 | 85.78 | 96.56 |
| | Energy fine-tuning (ours) | **51.23** | **89.71** | **97.63** |
| **LSUN-C** | Softmax score [14] | 66.54 | 83.79 | 96.35 |
| | Energy score (ours) | 35.32 | 93.53 | 98.62 |
| | ODIN [24] | 55.55 | 87.73 | 97.22 |
| | Mahalanobis [23] | 91.18 | 69.69 | 92.27 |
| | OE [15] | 21.92 | 95.81 | 99.08 |
| | Energy fine-tuning (ours) | **16.04** | **96.97** | **99.34** |
| **LSUN Resize** | Softmax score [14] | 82.42 | 75.38 | 94.06 |
| | Energy score (ours) | 79.47 | 79.23 | 94.96 |
| | ODIN [24] | 71.96 | 81.82 | 95.65 |
| | Mahalanobis [23] | **21.23** | **96.00** | **99.13** |
| | OE [15] | 69.36 | 79.71 | 94.92 |
| | Energy fine-tuning (ours) | 64.83 | 81.95 | 95.25 |
| **iSUN** | Softmax score [14] | 82.80 | 75.46 | 94.06 |
| | Energy score (ours) | 81.04 | 78.91 | 94.91 |
| | ODIN [24] | 68.51 | 82.69 | 95.80 |
| | Mahalanobis [23] | **26.10** | **94.58** | **98.72** |
| | OE [15] | 72.39 | 78.61 | 94.58 |
| | Energy fine-tuning (ours) | 67.23 | 79.36 | 94.37 |

Table 5: OOD Detection performance of CIFAR-100 as in-distribution for each specific dataset. The Mahalanobis scores are calculated from the features of the second-to-last layer. **Bold** numbers are superior results.

**Energy Bound Parameters** The optimal $m_{\text{in}}$ is $-23$ for CIFAR-10 and $-27$ for CIFAR-100. The optimal $m_{\text{out}}$ is $-5$ for both CIFAR-10 and CIFAR-100.

| $\mathcal{D}_{in}^{test}$ | fine-tune? | OOD dataset $\mathcal{D}_{out}^{test}$ | FPR95 ↓ | AUROC ↑ | AUPR ↑ |
|---|---|---|---|---|---|
| | | | Softmax score [14] / Energy score (ours) | | |
| **WideResNet** SVHN | ✗ | iSUN | 17.63 / **8.30** | 97.27 / **98.26** | 99.47 / **99.66** |
| | | Places365 | 19.26 / **9.55** | 97.02 / **98.15** | 99.40 / **99.63** |
| | | Texture | 24.32 / **17.92** | 95.64 / **96.17** | 98.96 / **99.00** |
| | | CIFAR-10 | 18.77 / **9.13** | 97.10 / **98.23** | 99.43 / **99.65** |
| | | LSUN-Crop | 31.60 / **26.02** | 94.40 / **94.59** | **98.79** / 98.75 |
| | | LSUN-Resize | 23.57 / **12.03** | 96.55 / **97.69** | 99.32 / **99.54** |
| | | **average** | 22.52 / **13.83** | 96.33 / **97.18** | 99.23 / **99.37** |
| | | | OE fine-tune [15] / Energy fine-tune (ours) | | |
| **WideResNet** SVHN | ✓ | iSUN | 0.56 / **0.01** | 99.82 / **99.99** | 99.96 / **100.00** |
| | | Places365 | 2.65 / **0.36** | 99.43 / **99.88** | 99.89 / **99.97** |
| | | Texture | 7.29 / **3.89** | 98.60 / **99.20** | 99.69 / **99.82** |
| | | CIFAR-10 | 2.14 / **0.17** | 99.50 / **99.90** | 99.90 / **99.98** |
| | | LSUN-Crop | 10.93 / **10.26** | 97.96 / 97.82 | **99.56** / 99.46 |
| | | LSUN-Resize | 0.63 / **0.00** | 99.82 / **99.99** | 99.96 / **100.00** |
| | | **average** | 4.03 / **2.45** | 99.19 / **99.46** | 99.83 / **99.87** |

Table 6: OOD detection performance comparison using softmax-based vs. energy-based approaches. We use WideResNet [43] to train on the in-distribution dataset SVHN with its training set only. We show results for both using the pretrained model (top) and applying fine-tuning (bottom). All values are percentages. ↑ indicates larger values are better, and ↓ indicates smaller values are better. **Bold** numbers are superior results.

(a) Effect of temperature $T$

(b) Effect of margin parameters

Figure 4: (a) We show the effect of $T$ on a CIFAR-10 pre-trained WideResNet. The FPR (at 95% TPR) increases with larger $T$. (b) Effect of margin parameters $m_{in}$ and $m_{out}$ during energy fine-tuning (WideResNet). The x-axes are on a log scale.