[Reviews · NeurIPS 2020]

Review 1

Summary and Contributions: This paper contributes a new algorithm for out-of-distribution detection based on energy scores extracted from a classifier trained in a purely discriminative way. The approach also includes an additional energy-bounded learning objective to increase the gap between in- and out-of-distribution data. --- UPDATE: Thank you for the rebuttal. I remain convinced the paper to be worthy of acceptance, and I appreciate that concerns raised by the reviewers will be taken into account to make the revision stronger.

Strengths: In my opinion, the main strengths of this work twofold: 1) The method is simple to understand and implement. This makes it very appealing, in particular as the energy scores can be extracted readily from pre-trained classifiers. 2) The extensive empirical study demonstrates the effectiveness of the method against competitive approaches. The discussion structured as well-defined research questions is quite helpful. Results are unanimously better than [13], [14], [20], [21], [15] and [7].

Weaknesses: - I am unfamiliar with this literature, but I find the setup of Section 3.2 a bit artificial. Is it realistic to assume we have out-of-distribution data to optimize the L_energy loss? Does considering one source of OOD data also help with respect to a second distribution of OOD data? - The experimental validation only considers CIFAR-10/100 as in-distribution data. While results are strong in this case, it is not clear to me if conclusions would generalize to other in-distribution datasets or if some should be nuanced. NB: If this is customary, then consider my remarks are genuine questions, rather than as weaknesses of your work.

Correctness: From what I understand, claims are all correct and substantiated.

Clarity: The paper is well structured and easy to follow. The manuscript offers a nice balance between a formal presentation of the method and its empirical study and validation from different viewpoints.

Relation to Prior Work: The prior work is clearly discussed. I appreciate the discussion with respect to [10], as it bears many similarities with this paper.

Reproducibility: Yes

Additional Feedback: Regarding the empirical methodology, in the case of energy fine-tuning, do you also consider a train/test split of the OOD data? (I suppose so, but couldn't find this information in the paper).


Review 2

Summary and Contributions: This paper presents an OOD detection method. It starts by showing how an EBM can be obtained from a pre-trained classifier and proposes to use the energy function as an OOD score. It then discusses why one should prefer the energy function over the commonly-used prediction confidence (i.e. the MSP). An additional component to the method involves training against a large dataset of outliers with a margin loss. Finally, the experiment section compares the proposed OOD detector to the MSP baseline and other OOD detection methods, including several methods that use generative training. The proposed OOD detector outperforms prior methods in most cases.

Strengths: The theoretical motivation for the energy function over the MSP is new and insightful (line 103). The comparison to discriminative and generative OOD detection methods is good, and the results improve over the SOTA method of Outlier Exposure when an outlier dataset is available to train against. Unlike several published OOD detection works, this paper uses sound methodology and does not select parameters on the OOD test data.

Weaknesses: Considering the similarities of the proposed method to JEM [10] and OE [14], its novelty is questionable. However, the JEM paper does not extensively evaluate their EBM on OOD detection, and their log p(x) results are much worse than this paper's results using a very similar method. If an explanation of this discrepancy is added, this could become a strength of the paper.

Correctness: The methodology is correct.

Clarity: The paper is clearly-written and well-structured.

Relation to Prior Work: The margin loss proposed in Section 3.2 is very similar to the method of Outlier Exposure [14]. Indeed, the PixelCNN++ experiments in [14] use a margin loss. This could be discussed more. Grathwohl et al. [10] demonstrate how log p(x) from an EBM can be used for OOD detection. For the purposes of OOD detection, this is identical to using the unnormalized energy function as proposed in this work. This could be discussed more.

Reproducibility: Yes

Additional Feedback: Regarding the relation of the proposed method to JEM [10], in Table 3 you compare to the version of JEM that uses the MSP rather than log p(x). In the JEM paper, the log p(x) detector gets much worse results than MSP applied to their model. However, the log p(x) detector is just the energy function shifted by the log of the partition function, which is a constant. Why is the energy function so much more effective in your experiments than the constant-shifted energy function in the JEM paper's experiments? Having two parameters in the margin loss seems redundant. What is the reasoning behind this design choice? Results on an additional, non-CIFAR in-distribution dataset would strengthen the paper. If my concerns are addressed, I will raise my score. _______________________________ Update after author feedback: I'm still not convinced the margin loss with two parameters is necessary, but if it consistently helps preserve classification accuracy on the in-distribution dataset then that's a reason to keep it. All my other concerns were addressed, so I'll raise my score to a 6.


Review 3

Summary and Contributions: The paper proposes replacing the overconfident softmax score with an energy-based one for the task of out-of-distribution detection. The main advantages of the approach are its applicability to already trained models, and robustness wrt hyperparameter tuning while achieving comparable/favorable results against baselines. The experiments and results are promising and thorough, however, the presentation of the method can be improved.

Strengths: 1. The presented results motivate the preference of energy-based score over softmax through analysis and empirical evaluation. 2. Appropriate comparison against discriminative and generative energy-based methods.

Weaknesses: 1. A paragraph/section on 'Problem setting' is missing (no notation, variables introduced), the definiton of the energy function itself is not clear, is it E(\dot, \dot) or E(\dot)? Is y missing? What is the difference between y and y'? I think the whole section 2 can be rewritten in a more clear way (better flow and references from text books, relevant papers). 2. The key observation, starting at line 92 can be made more formal, or maybe show an syntetic experimet for validation of the claim. 3. Recent work (https://arxiv.org/abs/1906.02845) show that denisty based models fail at OOD, I wonder if the authors have tried similar experiments and if the same issue arises with the energy-based score (since it should be aligned with the in-domain density). If OOD from more complex (CIFAR/SVHN) -> simpler datasets (MNIST) works this will show one more advantage of the energy-score. 4. Were other architectures explored besides WideResNet? 4. a. Maybe add a toy example on synthetic data (such as two concentric rings or moons dataset) to showcase the method works with a simple network as well. 5. Details in Appendix B state that results are averaged over 10 runs. Why is there no indication of standard deviation? Were these 10 runs done over random hyper-paramter configuration for all baselines or 10 runs with the best selected hyperparameters' values?

Correctness: Section 2 needs to be improved.

Clarity: In my opinion the paper can benefit a lot from improved presentation. Especially sections 2 and 3. Please see more details under [Weaknesses] above.

Relation to Prior Work: The authors did good job in the experimental section where they show the advantages of their method over JEM which seems to be the closest baseline. However, I am not convinced about methodological distinciton between JEM and the Energy score one, would help if this is more elaborated in Section 3.

Reproducibility: Yes

Additional Feedback: Minor comments: - What exactly do you mean by "unified" framework? - In Table 1, AUCROC arrow should point upwards. - Maybe naming the method would help the presentation/distinciton from existing approches ------------------------------------------------------------ Update after Author Feedback and Discussion ------------------------------------------------------------ Thank you to the authors for their detailed feedback. My questions and concerns have been mostly addressed/answered. I would strongly advise for revision and rewriting of Sec. 2 and 3.


Review 4

Summary and Contributions: This paper shows that energy scores are better to distinguish in and out-of-distribution samples than the softmax scores. It also shows the mathematical connection between the energy score and the softmax confidence score. The energy score is well aligned with density p(x). According to experimental results, energy-based method outperforms the state-of-the-art on common benchmarks.

Strengths: Clear analysis on why the energy score is better than the softmax scores for out-of-distribution detection. The mathematical connection between the energy score and the softmax confidence score in the paper is clear. And the experimental results in Table 1 show that using energy scores reduces the average false positive rate. It is good that the paper also show the in-distribution performance for the method in Table 2, not only out-of-distribution performance. The method show the robustness.

Weaknesses: Comparing this working [10] in the paper, the fine-tuned energy achieves better results with the hinge-loss training. It is great if more coparisions of these two training methods could be shown. The new training method seems help a lot for the out-of-distribution detection. And it is an important distributions comparing to [10]. [10]: Will Grathwohl, Kuan-Chieh Wang, Joern-Henrik Jacobsen, David Duvenaud, Mohammad Norouzi, and Kevin Swersky. Your classifier is secretly an energy based model and you should treat it like one.

Correctness: I just realized a key point. It is true that: -E(x,f) proportionally to log p(x) when f is fixed, however f is finetuend. It is hard to compare the unnormalized energy scores between f and finetuned f. I think the necesssay conditioan is f is self-normalized (if f_y(x) is the log score correspong to the y-th label, not the logit). In Equation (5), the author use the unnormalized energy score for OOD dection. I think the situation hold when f is self-normalized as mention above. The author could check whether it could work well when using log score instead logits. I guess this the reaon there are square loss in Equatioin (7) and (8). The sum of square loss have the effect to bound the energy score. It has the effect of self-normalization for f.

Clarity: Yes.

Relation to Prior Work: There are some related work on margin-based training for energy-based model.

Reproducibility: Yes

Additional Feedback: In Table 1, it should be uparrow for AUROC.

[Author Response · NeurIPS 2020]

We thank the reviewers for their insightful feedback. We are encouraged that they find our motivation and idea to
be appealing (**R1**), theoretically insightful (**R2**), new (**R2**) and clearly presented (**R2**, **R4**). We are equally glad they
found our results strong (**R1**), sound (**R2**), promising and thorough (**R3**), appropriately positioned w.r.t. prior work
(**R1**, **R3**), and outperforming the state-of-the-art (**R2**, **R4**). Moreover, we are pleased that **R2** and **R4** appreciated our
mathematical insights on why the energy score is better than the softmax score for out-of-distribution detection. We
address reviewer comments below and will incorporate all feedback.

[**R1**, **R2**] **Can you evaluate on additional in-distribution dataset?** We have run experiments using SVHN as in-
distribution dataset. Using energy score consistently outperforms softmax score by **8.56**% (FPR at 95% TPR), which
suggests its general applicability beyond CIFAR datasets. We will definitely include these results in the final version.

[**R2**, **R3**] **How to distinguish methodological differences w.r.t. prior work?** As discussed in L240–L248, there are
several major distinctions. We have moved the expanded discussions to Section 3 for clarity, as **R3** suggested. To recap:

• The idea of using the energy score for OOD detection is novel and theoretically motivated. JEM [10] used
likelihood score $\log p(\mathbf{x})$ from **generative** modeling or softmax score; we instead derive the energy score from a
**pure discriminative** model. We showed both mathematical insights (L103–L117) and empirical success that energy
is better than softmax score—which was not presented in OE [14] or JEM [10].
• JEM's optimization is intractable as it requires estimating the normalized densities w.r.t. all inputs $\mathbf{x}$. Ours is easy to
optimize using standard SGD, and doesn't involve proper normalization.
• JEM only uses in-distribution data. Our training objective (6) leverages auxiliary outlier data and explicitly enlarges
the energy gap, which is also different from OE.

[**R1**] **Is it realistic to assume we have OOD data to optimize the objective? Does one source of OOD data also**
**help another?** Thank you for the question! In fact, we are not the first to use auxiliary data for OOD detection. Previous
work (Outlier Exposure [14]) showed one can leverage a large, diverse auxiliary outlier dataset for better estimation on
the decision boundary for OOD detection. Since the auxiliary data (i.e., 80 million TinyImages) is sufficiently diverse,
this fine-tuning process helps generalize to different test-time OOD data, as evidenced in Table 1.

[**R1**] **Is there a train/test split of the OOD data for fine-tuning?** Following common practice, we used 80M
TinyImages as auxiliary OOD training data; we tested on six OOD datasets (L143) that are different from training.

[**R2**] **Why are the $\log p(x)$ results in JEM [10] much worse?** Very insightful question! While the energy score can
be interpreted from the likelihood view, the optimization processes of JEM vs. ours are very different. We believe
the difficulty in optimization led to the worse results we see in JEM. Specifically, JEM's objective estimates the
joint distribution $\log p_\theta(\mathbf{x}, y) = \log p_\theta(\mathbf{x}) + \log p_\theta(y|\mathbf{x})$. The first term maximizes the likelihood from a generative
modeling perspective, which requires estimating normalized densities. As pointed out by Grathwohl et al. [10] in the
limitations (Sec. 6), *"Energy based models can be very challenging to work with. Since normalized likelihoods cannot*
*be computed, it can be hard to verify that learning is taking place at all … Furthermore, the gradient estimators we*
*use to train JEM are quite unstable and are prone to diverging if the sampling and optimization parameters are not*
*tuned correctly"*. In contrast, our training objective is purely discriminative and can be more easily optimized using
standard SGD.

[**R2**] **Why having two margin loss parameters?** One problem with the single margin is that the difference of the
energy scores between in- and out-of-distribution is relatively fixed; however, the energy scores can change over the
training process. This results in lower accuracy of in-distribution classification (-0.57% for CIFAR-10 and -1.11% for
CIFAR-100) because the training is less stable due to the drift of energy scores from batch to batch. Instead, using two
margin hyper-parameters results in overall good performance on both in-distribution classification and OOD detection.

[**R3**] **Does the method work on more complex (CIFAR/SVHN) → simpler dataset (MNIST)?** Yes. On the CIFAR-
10-pretrained WideResNet, using the energy score yields **14.50**% FPR when evaluated against MNIST as OOD data,
outperforming 51.73% using the softmax score.

[**R3**, **R4**] **Other architectures/toy example/more comparison between energy and JEM:** Yes, consistent improve-
ment was also observed on other architectures such as AllConv and DenseNet. We will include these results and the
suggested toy example, more comparisons with JEM in the final version.

[**R3**] **Were these 10 runs done over random hyper-parameter configurations or 10 runs with the best selected**
**hyperparameters' values?** The latter. We will make sure to report variances in the final version.

[**R4**] **What if you use self-normalized output, replacing logits with log score?** We found the suggestion of self-
normalization very interesting, and investigated further. Mathematically, the log score is equivalent to logit shifted by the
free energy, which is *sample specific* and therefore might change the overall score distribution. This shifting in actuality
leads to the free energy score (on self-normalized output) collapsing to 0, i.e., $\text{LogSumExp}(\log\ softmax_i) = 0$, and
did not work well. We believe the normalizing factor needs to be *sample independent* to ensure $E(\mathbf{x}, f)$ is proportional
to the density. We are excited to explore this as part of future work.

[**R2**, **R3**, **R4**] **Typo/Writing clarity:** All fixed. We thank the reviewers for the careful read and helpful suggestions!

[Meta-Review · NeurIPS 2020]

All four referees support accept. As pointed out by reviewers, AC also agree that the proposed method is easy to use and the experimental results are good (but, more architectures and non-vision datasets are encouraged to test). As also pointed out by a reviewer, JEM [10] also uses an energy-based modeling. AC also thinks that Mahalanobis [20] also uses a type of energy-based modeling. Hence, AC suggest the authors to highlight the advantage of the proposed scheme over [10] and [20] in the final draft (they did some in the rebuttal letter though). Irrespectively, AC recommend acceptance, as the proposed scheme is easier to use than [10] and [20].